# Coupling of HSP72 α-Helix Subdomains by the Unexpected Irreversible Targeting of Lysine-56 over Cysteine-17; Coevolution of Covalent Bonding

**DOI:** 10.3390/molecules25184239

**Published:** 2020-09-16

**Authors:** Aimen Aljoundi, Ahmed El Rashedy, Patrick Appiah-Kubi, Mahmoud E. S. Soliman

**Affiliations:** Molecular Bio-Computation & Drug Design Lab, School of Health Sciences, University of KwaZulu-Natal, Westville, Durban 4000, South Africa; aymanaljundi69@gmail.com (A.A.); ahmedelrashedy45@gmail.com (A.E.R.); appiahpat@gmail.com (P.A.-K.)

**Keywords:** covalent MD simulation, HSP72, 8-*N*-benzyladenosine, coupling, principal component analysis

## Abstract

Covalent inhibition has recently gained a resurgence of interest in several drug discovery areas. The expansion of this approach is based on evidence elucidating the selectivity and potency of covalent inhibitors when bound to particular amino acids of a biological target. The unexpected covalent inhibition of heat shock protein 72 (HSP72) by covalently targeting Lys-56 instead of Cys-17 was an interesting observation. However, the structural basis and conformational changes associated with this preferential coupling to Lys-56 over Cys-17 remain unclear. To resolve this mystery, we employed structural and dynamic analyses to investigate the structural basis and conformational dynamics associated with the unexpected covalent inhibition. Our analyses reveal that the coupling of the irreversible inhibitor to Lys-56 is intrinsically less dynamic than Cys-17. Conformational dynamics analyses further reveal that the coupling of the inhibitor to Lys-56 induced a closed conformation of the nucleotide-binding subdomain (NBD) α-helices, in contrast, an open conformation was observed in the case of Cys-17. The closed conformation maintained the crucial salt-bridge between Glu-268 and Lys-56 residues, which strengthens the interaction affinity of the inhibitor nearly identical to adenosine triphosphate (ADP/Pi) bound to the HSP72-NBD. The outcome of this report provides a substantial shift in the conventional direction for the design of more potent covalent inhibitors.

## 1. Introduction

Heat shock proteins (HSPs) play a central role in the clearance of damaged proteins by inducing protein aggregation and proteotoxicity. This process occurs by preventing inappropriate stress-induced protein aggregation, ensure proper refolding of denatured proteins, and, if necessary, the promotion of their degradation [1,2,3]. Recent studies have proven that increased protein synthesis (translation) is vital to the conversion of neoplasms. As a result of this increase, cancer cells appear to be particularly susceptible to agents that inhibit the removal of aggregated or misfolded proteins generated by protein synthesis as a product [4,5,6]. Hsp70 protein member families are among the highly conserved proteins and play a critical role in these processes [7]. The primary stress-inducing member of the Hsp70 chaperone family is known as Hsp72 and is encoded by two genes, HSPA1A and HSPA1B, which generate isoforms of Hsp72 [8]. Hsp72 is extremely homologous to the 78 kDa glucose-regulated protein, which plays a significant role in organizing the unfolding protein response [9]. Hsp72 is expressed at high levels in malignant tumors of various origins [10] and enhances cancer cell survival [11,12]. Thus, the inhibition of Hsp72 is considered to be a successful pathway in anti-tumor therapy [13]. 

All the different functions of Hsp70s are accomplished through a transient chaperone interaction with substrate proteins through its C-terminal substrate-binding domain (SBD) [14]. The nucleotide binds allosterically to the *N*-terminal nucleotide-binding domain (NBD) to control the transient chaperone interaction. The affinity of the SBD for substrates decreases by 10- to 400-fold when ATP is binding to the NBD. Hence, the inhibition of NBD is considered one of the most promising strategies for HSP72 function inhibition [15]. The NBD consists of two adjacent lobes (lobe I and lobe II), which form a deep nucleotide groove connected to the base. Each lobe consists of two subdomains (IA, IIA, IB, and IIB) [16,17]. Domains IB and IIB are linked to IA and IIA, respectively, by flexible hinges and control access to the nucleotide-binding sites [18] (Figure 1).

Several studies have designed potential Hsp72 inhibitors, including 2-phenylethynesulfonamide (PES) [19], 15-deoxyspergualin (DSG) [20], natural products Oridonin [21] and Novolactone [22], but upregulation is one of the most challenges associated with drug resistance and poor clinical outcomes [23]. The challenging hurdle to cellular activity for competitive nucleotide inhibitors of HSP72 is due to the highly conserved domain. This conserved domain is mostly occupied by ADP and ATP (ADP, KD ~ 110 nm) in addition to hydrophilic and electrostatic interactions with the nucleotide ribose and phosphate amino acid residues, hence difficult drug binders [24]. Covalent inhibition is a key approach for high-affinity proteins [25] and has recently sparked interest among the community of pharmaceutical research [26]. Covalent inhibition occurs when the electrophilic moiety of a covalent ligand connects with a nucleophilic residue of a biological target, resulting in an irreversible link between the protein and the drug [8]. For example, it can inhibit the same biological target at a lower concentration than a noncovalent drug due to the long-lasting effects of a covalent drug [27,28]. An example of a covalent reaction between a ligand and its protein target is shown in Figure 2.

In a recent study by Pettinger et al. (2017) using fluorescence polarization (FP) assay and crystallography, the authors observed an unexpected covalent bond interaction between 8-N-benzyladenosine and lysine-56 of the NBD of HSP72 (HSP72-NBD domain). This unexpected covalent bond interaction resulted in the arrest of the NBD via hydrogen-bonding array of the ribose and adenine moieties with the lipophilic para-chlorobenzylamine moiety, parallel with the two α-helices of the binding cavity [29] (Figure 3). It is worth highlighting that the observed covalent inhibition of HSP72 via lysine-56 by 8-*N*-benzyladenosine was opposed to the anticipated 8-*N*-benzyladenosine covalent inhibition of HSP72 via cysteine-17.

The unexpected preferential covalent bond formation of 8-*N*-benzyladenosine with lysine-56 over cysteine-17 prompted the need to investigate the conformational plasticity and structural dynamics associated with this unexpected covalent interaction. To accomplish this, we utilized in silico approaches such as covalent molecular dynamics simulation, clustering and principal component analyses to define and compare the structural dynamics of 8-*N*-benzyladenosine-Lys-56 modeled covalent complex with 8-*N*-benzyladenosine-cys-17 covalent complex on HSP72-NBD domains. 

Extensive analyses reveal that the coupling of the inhibitor to cysteine-17 is intrinsically more dynamic than to lysine-56, mainly in the IIA and IA α-helices region. Conformational dynamics analysis further reveals that the coupling of 8-*N*-benzyladenosine to lysine-56 induces a closed conformation of the IIB and IIA α-helices of the NBD. In contrast, an open conformation was observed when coupled to the cysteine-17 residue. 

## 2. Computational Methodology 

### 2.1. System Preparation 

The studied models were prepared based on the reported human HSP72-NBD domain crystal structure retrieved from the Protein Data Bank [30] (PDB code: 5MKS) [31] and set up using the UCSF Chimera software package [32]. The ligand was prepared using MarvinSketch 6.2.1, 2014, Molegro Molecular Viewer (MMV), and Chem-Axon (http://www.chemaxon.com) to ensure that the ligand hybridization state and proper angels were displayed [33,34]. 

### 2.2. Covalent Docking

Covalent bonds were formed between the inhibitor and lysine-56, cysteine-17 residues of HSP72-NBD. Before the creation of the covalent bond, a non-covalent docking was performed for the inhibitor to ensure an appropriate stable binding mode at HSP72-NBD. This initial non-covalent docking was performed using AutoDock Tools GUI, and AutoDock Vina [35] integrated with Chimera GUI. The AutoDock Tools GUI was used to define the grid box (center: −12.75, −8.433, and 5.19; Size: 18.65, 18.02, and 21.39) at the lysine and cysteine binding site of the protein. The initial non-covalent docking was to appropriately position the inhibitor in the active site to allow the covalent bond to be created subsequently. The binding poses were checked to see if the two atoms that will eventually form the covalent bond between the target residues (lysine-56 and cysteine-17) and the inhibitor was within 3 Å of each other. The inhibitor binding mode that did not meet this condition to allow the bond to be formed was rejected. The Schrödinger Maestro [36] was used to create the covalent bond between the inhibitor and the residues (lysine-56 and cysteine-17). The best covalent complex pose was then selected.

The protein preparation wizard in Maestro Schrödinger was used to optimize the protonation state of the complex, adjust hydrogen atoms, cap acetyl, and methylamide neutral residues. An initial vacuum minimization was performed to resolve any steric clashes and restore normal bond lengths. The prepared receptor and covalent ligand were saved separately and taken for parameterization using the preparatory program. The APO system was run based on our previously reported protocol for non-covalent simulations [37,38].

### 2.3. Covalent Molecular Dynamic Simulation

The two covalent systems were exposed to an all-atom classical covalent molecular dynamics simulation (MD) using the PMEMD package in Amber 14 [39]. The Antechamber module in the Amber 14 was used to provide atom types and atomic ligand partial charges using the FF14SB forcefield [40]. The LEaP program was used to generate a library defining the ligand residue topology. The final system was built, neutralized, and solvated with two Na^+^ counter-ions using the Dabble program [41]. Before starting the simulation process, the studied covalent complexes were placed within a box of TIP3P water molecules with 10 Å distance from the protein [42]. Particle mesh Ewald (PME) method was implemented within Amber14, with direct space and Van der Waals cut-off of 12 Å, to obtain long-range electrostatic interaction. To further relax the complex and remove potential steric clashes, each system was energy minimized for a total of 7500 steps (2500 steps of steepest descent and 5000 conjugate gradient steps) with a 10 kcal/mol/Å^2^ restraint conditions applied. The systems were heated for 30 ps from 0 to 300 K with an additional 7-ns equilibration performed at a 4-fs integration time step. MD simulation production runs of 250 ns were performed for each system during which the SHAKE algorithm was used to constrict all atomic hydrogen bonds at a 4-fs integration time step [43]. The computational methodology concerning the covalent systems was based on our previously reported [44]. The CPPTRAJ and PTRAJ modules [45] of AMBER14 package were used to analyze resulting trajectories for root mean square deviation (RMSD), root mean square fluctuation (RMSF), solvent accessible surface area (SASA), and secondary structure analysis. The data were expressed in mean ± standard deviation. The obtained data were plotted using Microcal Origin tools [46] and Maestro Schrödinger software [36].

### 2.4. Clustering and Principal Component Analysis

A principal component analysis (PCA) was performed to describe the internal motion of the complexes using the Bio3D package in R. The process involves the initial construction of the covariance matrix (C) from (x,y,z) coordinate positions of the C-atoms as representatives of residues (N), generating a large matrix of dimension 3N_3N. The covariance matrix was further diagonalized to obtain eigenvectors based on related eigenvalues. This was then projected on the first three eigenvectors (PC1, PC2, and PC3). 

## 3. Result and Discussion

To understand the molecular behavior associated with the preferential binding mechanism of 8-*N*-benzyladenosine to Lys-56 over Cys-17 at HPS72-NBD, covalent molecular dynamics simulation was employed to study the structural and dynamical changes of the above two covalent binding models. Post molecular dynamics analyses were carried out covering different aspects, including dynamic conformational stability (RMSD, RMSF), dynamic system variations (SASA, secondary structure analysis, PCA).

### 3.1. Overall Structural Stability and Dynamics of the Simulated Systems 

To assess the structural stability of the studied systems, the root mean square deviation (RMSD) was calculated based on C-α atoms for the Apo and covalent complexes over the 250ns simulation. The systems achieved stable equilibration after 50ns. The recorded average RMSD values for the entire frames of the systems were 2.1 ± 0.27 Å, 3.1 ± 0.42 Å, and 4.8 ± 0.54 Å for Apo-NBD, Lys-NBD, and Cys-NBD, respectively (Appendix A). The Apo system showed a lower RMSD average value, whereas the Lys-NBD induces a relatively more stable protein conformation compared with the Cys-NBD complex.

The covalent binding effects of 8-*N*-benzyladenosine towards amino acid residues of the HSP72-NBD when covalently bonded to Lys-56 or Cys-17 were analyzed by the root mean square fluctuation (RMSF). The mobility of HSP72-NBD C-α RMSF was computed and averaged over 250ns to observe inhibitor binding effects towards HSP72-NBD protein structural flexibility. The computed average atomic fluctuations for Apo-NBD, Lys-NBD, and Cys-NBD were 1.2 ± 0.52 Å, 1.6 ± 0.77 Å, and 2.1 ± 0.94 Å, respectively. The plotted results in Figure 4 indicate overall relative lower residue fluctuation in the Lys-NBD complex system compared with the Cys-NBD complex system. This observation suggests that the covalent binding of 8-*N*-benzyladenosine to HSP72-NBD via lysine relatively decreases the overall protein flexibility in contrast to 8-*N*-benzyladenosine binding to HSP72-NBD via cysteine residue.

Furthermore, to obtain insight into how the protein surface interacts with solvent molecules and how it relates to the compactness of the hydrophobic protein core, the solvent-accessible surface area (SASA) of the protein upon ligand binding was calculated (Figure 5). This was accomplished by calculating the surface area of the protein visible to solvent across the 250 ns MD simulation, which is vital for biomolecular stability [47]. The overall SASA indicates that the Lys-NBD protein surface is relatively less exposed to solvent molecules compared with Cys-NBD inhibition. The computed average SASA values for Apo-NBD, Lys-NBD, and Cys-NBD inhibition systems were 17374.3 ± 327.7 Å, 17975.5 ± 518.2 Å, and 18289.5 ± 680.2 Å, respectively. 

### 3.2. Analysis of Secondary Structure Variation 

To further gain an additional structural understanding of the cysteine-NBD targeting, and the lysine-NBD targeting of HSP72, the DSSP classification for each amino acid was calculated (Figure 6). The segment of residues 50–70, which was suggested in the stabilization of the closed formation [48,49,50], remains as α-helix-para-turn (green-blue-brown) conformation throughout the simulation when bound to cysteine. In contrast, it becomes 3_10_helices-bend (dark green-red) when bound to lysine is binding. The α-helix in this segment disappears almost entirely and thus potentially destroys the NBD as it is no longer in close proximity. The solvent-exposed salt-bridge between Glu-268 and Lys-56 is absent [51].

### 3.3. Conformational Clustering and Principal Component Analysis 

Principal component analysis and clustering were performed for Cys-17-HSP72 and Lys-56-HSP72 8-*N*-benzyladenosine covalent complexes to observe the overall concerted motion of HSP72-NBD protein. The PCA provides insight into the conformational changes of macromolecules, such as proteins [52]. The structural distribution of conformational changes and the proportion of variance of the captured eigenvectors are shown in Figure 7A,C. The first three principal components accounted for 67.8% and 53.2% of the total variance observed in the MD trajectories for Cys-17-HSP72 and Lys-56-HSP72 covalent complexes, respectively. The magnitude of principal component 1 (PC1) was observed to be the highest for the Cys-17-HSP72 complex (51.2%); however, a relatively lower PC1 of 30.8% was observed for Lys-56-HSP72 covalent complex. The observed PC1 variance suggests that the inhibitor-induced radical conformational changes in the protein structure when couple to Cys-17, resulting in a dynamic rearrangement of the IIA and IA helices of HSP72-NBD (Figure 7B). The clustering analysis also shows conformational distribution variance along the first, second, and third principal components with each dot representing a single complex conformation. The clustering and principal component analyses suggest that HSP72 undergoes a large conformational change in the NBD when the inhibitor is coupled to Cys-17 compared with Lys-56 (Figure 7). The coupling of the inhibitor to cysteine-17, therefore, shows intrinsically more dynamic residue mobility largely in the IIA α-helix region compared with the lysine-56 inhibitor complex (Figure 7B,D).

### 3.4. Understanding Structural Dynamics upon 8-N-Benzyladenosine Coupling to Lysine and Cysteine of HSP72-NBD α-Helices

To further elucidate the structural basis of the preferential covalent coupling of 8-*N*-benzyladenosine with Lys-56 over Cys-17, we compared the subdomain helix dynamic motions throughout the simulation. The observed structural changes in the helix motion show a different mechanism of the NBD characterized by an opened conformation and a closed conformation of the IIB and IIA subdomains for Cys-17 HSP72 complex and Lys-56 HSP72 complex, respectively (Figure 8). The Lys-56-HSP72 8-*N*-benzyladenosine complex binding results in the closure of the two helices for the subdomain IIB and IIA, thus, inducing a closed conformation. On the other hand, an open conformation of the two helices for the subdomain IIB and IIA was observed for the Cys-17-HSP72 8-*N*-benzyladenosine complex.

The inhibition of HSP72-NBD via lysine-56 decreased the inter-residue distance between two opposite IIB and IIA subdomain helix residues Thr-265 and Asn-57 with an average distance of 9.24 Å. However, a distance of 26.14 Å was observed between residues Thr-265 and Asn-57 via the cysteine-17 inhibition model (Figure 9). The coupling of 8-*N*-benzyladenosine to Lys-56 over Cys-17 residue strengthens the interaction affinity of 8-*N*-benzyladenosine and induced a closed conformation of the IIB and IIA nucleotide-binding subdomain. 

Further analysis of the 8-*N*-benzyladenosine interaction mechanism with Lys-56 and Cys-17 reveals a salt-bridge interaction between Lys-56 and Glu-268 in the Lys-56-HSP72-NBD complex (Figure 10A), which is absent in the Cys-17-HSP72-NBD complex (Figure 10B). The observed Lys-56-8-*N*-benzyladenosine interaction and the closed conformation is nearly identical to the closed conformation of ADP/Pi bound to HSP72-NDB [51]. In Cys-17 inhibitor coupling, the bulk of the inhibitor is sandwiched between Glu-268 and Lys-56 of the two helices, which prevent the possible formation of the Glu-268 and Lys-56 salt bridge inducing an open conformation (Figure 10B).

## 4. Conclusions

The human heat shock protein 72 (HSP72) represents a vital therapeutic target during the critical stages of oncogenesis and progression of human cancers. In this study, we performed covalent molecular dynamic simulation followed by extensive analyses to decipher the structural basis and conformational dynamics associated with the unexpected preferential coupling of 8-*N*-benzyladenosine to lysine-56 over cysteine-17 in HSP72-NBD models. The results reveal that the irreversible binding of 8-*N*-benzyladenosine to lysine-56 over cysteine-17 in HSP72-NBD represents the most stable conformation with minimal intrinsic dynamics. Clustering and PCA showed that the first three principal components accounted for 67.8% and 53.2% of the total variance for Cys-17-HSP72 and Lys-56-HSP72 covalent complexes, respectively. The magnitude of principal component 1 (PC1) was observed to be the highest for the Cys-17-HSP72 complex (51.2%); however, a relatively lower PC1 of 30.8% was observed for Lys-56-HSP72 covalent complex. 

The conformational dynamics analysis further reveals what the experimental study could not capture and explain, that the coupling of 8-*N*-benzyladenosine to Lysine-56 induces a closed conformation of the IIB and IIA α-helices of the nucleotide-binding subdomain. In contrast, an open conformation was observed in coupling to Cysteine-17 residue. Interestingly, the close conformation maintained the crucial salt-bridge between Glu-268 and Lys-56 residues, which strengthens the interaction affinity of 8-*N*-benzyladenosine nearly identical to ADP/P_i_ bound to the HSP72-NBD. It is rare for non-catalytic lysine residues to form a covalent bond with an inhibitor. The recent unexpected covalent formation between lysine-56 and 8-N-benzyladenosiness would assist with the design of more potent and highly selective covalent inhibitors for HSP72 with the potential to overcome drug resistance challenges and represent a novel therapeutic approach for inhibiting HSP72 oncoprotein. 

## Figures and Tables

**Figure 1 molecules-25-04239-f001:**
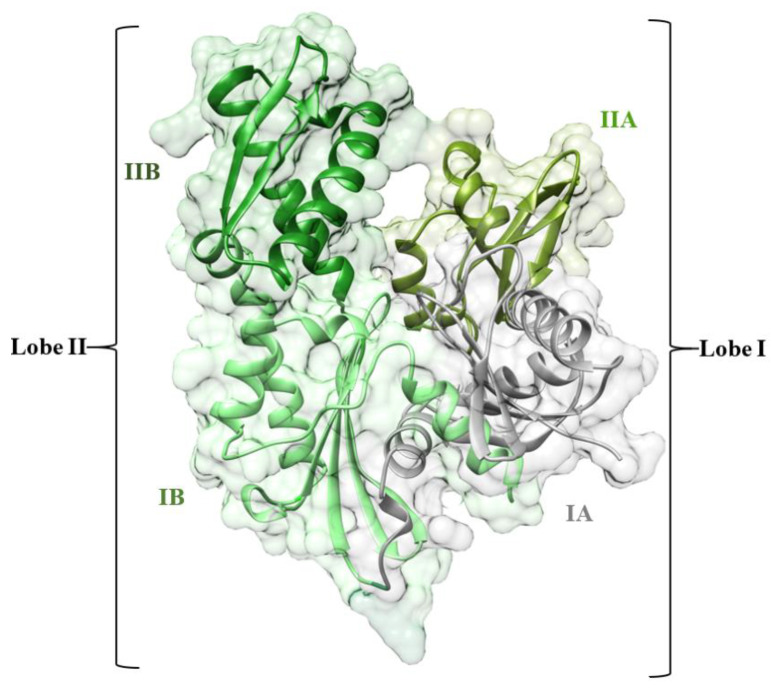
The 3-D crystal structure of the HSP72-NBD protein (PDB code: 5MKS). The IA, IIA, IB, and IIB subdomains are shown in green, light-green, oily green, and grey, respectively.

**Figure 2 molecules-25-04239-f002:**
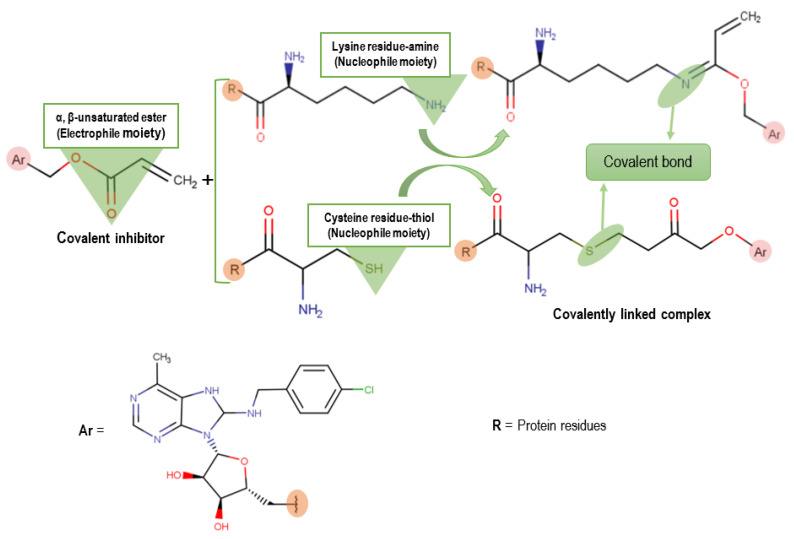
A schematic summarizing the covalent reaction mechanism between a covalent inhibitor and the protein residues lysine and cysteine.

**Figure 3 molecules-25-04239-f003:**
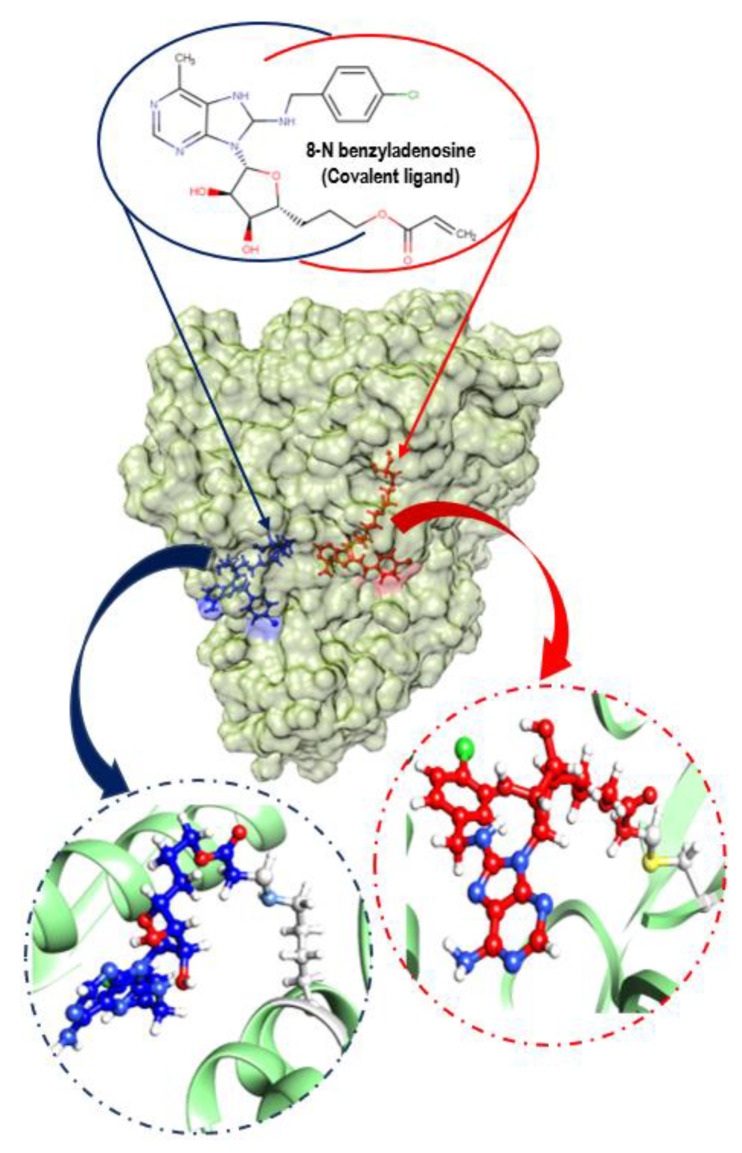
Surface view of HSP72 covalently bonded with 8-N-benzyladenosine inhibitor via a cysteine residue (red) and a lysine residue (navy-blue) with different binding pocket. A close view of two covalent bonds (yellow color cysteine residue bonding and light blue for lysine residue bonding).

**Figure 4 molecules-25-04239-f004:**
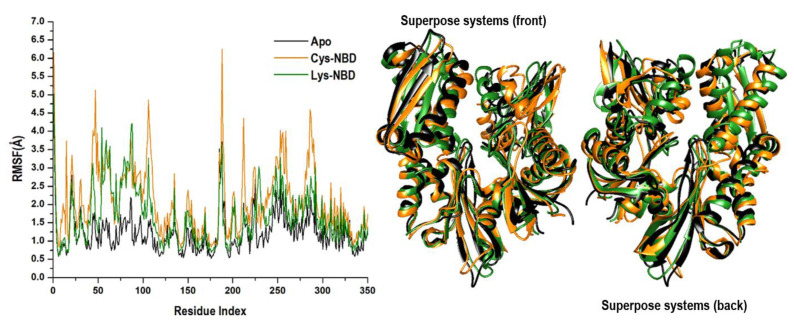
The time evolution RMSF of each residue of the protein C-α atoms over 250 ns for Apo (black color), and Cys-NBD (orange color), and Lys-NBD (green color). Superposed crystal structures of the studied systems are also illustrated to show differences in fluctuations.

**Figure 5 molecules-25-04239-f005:**
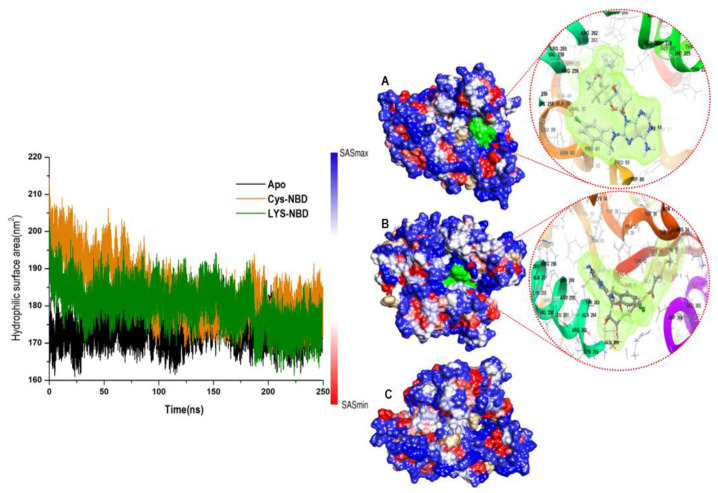
Solvent accessible surface area (SASA) backbone atoms relative to the starting minimized structure over 250 ns for Apo, Cys-NBD, and Lys-NBD. Areas with higher SASA values and lower SASA values are shown in blue and red, respectively, for Cys-NBD (**A**), and Lys-NBD (**B**) and Apo (**C**).

**Figure 6 molecules-25-04239-f006:**
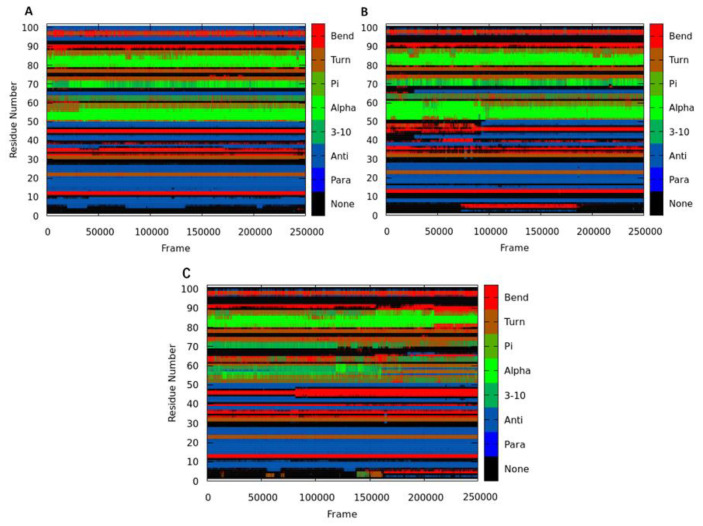
DSSP classification for the time evolution of the secondary structural elements for Apo (**A**), Cys-NBD (**B**), and Lys-NBD (**C**).

**Figure 7 molecules-25-04239-f007:**
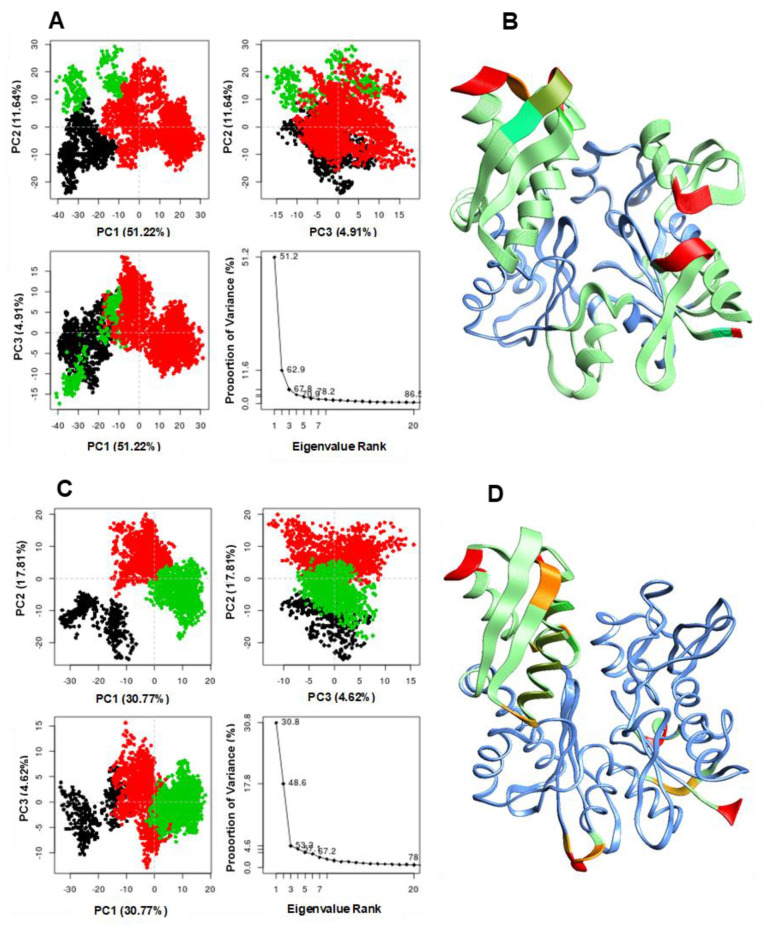
Clustering and principal component analysis on 250ns of equidistance conformations using the Bio3D package in R. The plots show the first three eigenvectors for Cys-17 HSP72 complex (**A**) and Lys-56 HSP72 complex (**C**) Conformers are colored according to the k-means clustering: cluster 1, black; 2, red; 3, green. dominant motions and captured eigenvector variance; and residue mobility for Cys-17 HSP72 complex (**B**) and Lys-56 sHSP72 complex (**D**). The color scale from blue, green, to red depicts low to high atomic displacements.

**Figure 8 molecules-25-04239-f008:**
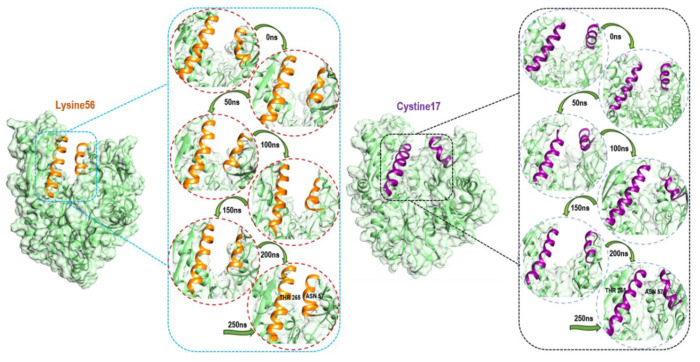
Conformational dynamic comparison of α-helix coupling upon 8-*N*-benzyladenosine binding to Lys56 (Orang color) and Cys17 (violet color) showing the closed conformation of HSP71-NBD.

**Figure 9 molecules-25-04239-f009:**
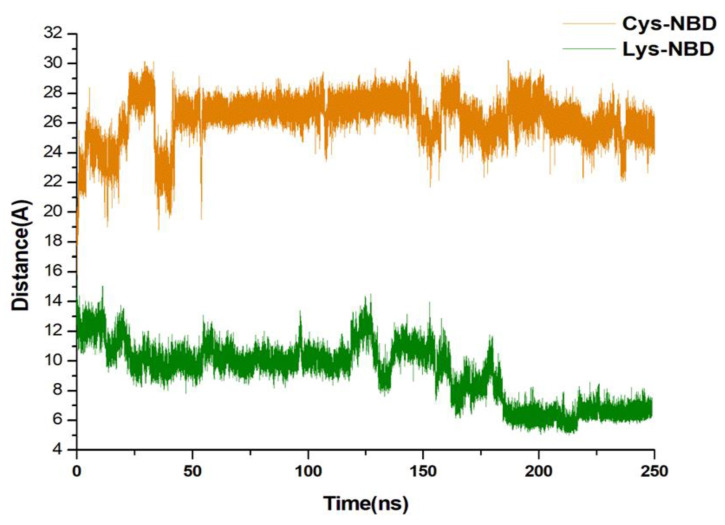
Distance between C- α residues involving THR265, and ASN57 α-helices of the NBD. The average distances were found to be 9.24 Å and 26.14 Å, respectively, for Lys-NBD and Cys-NBD conformations.

**Figure 10 molecules-25-04239-f010:**
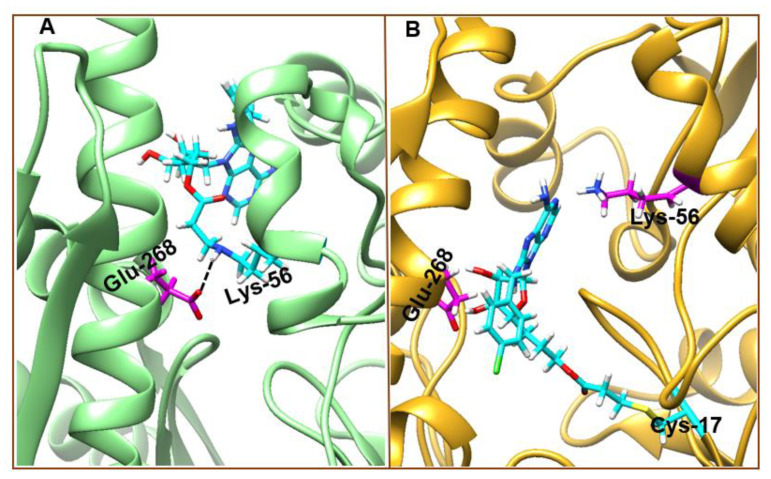
**s**. Lys-56-8-*N*-benzyladenosine (**A**) and Cys-17-8-*N*-benzyladenosine (**B**) coupled HSP72-NBD irreversible interaction mechanism. Crucial salt-bridge between Glu-268 and Lys-56 residues were missing in the Cys-17 coupled conformation.

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
