# Peer review of "Coupling of HSP72 α-Helix Subdomains by the Unexpected Irreversible Targeting of Lysine-56 over Cysteine-17; Coevolution of Covalent Bonding"

_molecules, 2020, doi:10.3390/molecules25184239_

Round 1

Reviewer 1 Report

Aljoundi et al. performed structural and dynamic analyses to investigate the mechanism of covalent inhibition of HSP72 by preferentially targeting Lys56 instead of Cys17. They conducted covalent molecular dynamics simulation, including RMSD, RMSF and SASA, clustering and principal component analyses revealing that model compound 8-N-benzyladenosine binding to HSP72-NBD lysine requires less energy and the product is more stable.

Overall, it's a nice paper. I have one tiny input. In figure 2, lysine residue-thiol need change to lysine residue-amine”.

Author Response

Q1. Aljoundi et al. performed structural and dynamic analyses to investigate the mechanism of covalent inhibition of HSP72 by preferentially targeting Lys56 instead of Cys17. They conducted covalent molecular dynamics simulation, including RMSD, RMSF and SASA, clustering and principal component analyses revealing that model compound 8-N-benzyladenosine binding to HSP72-NBD lysine requires less energy and the product is more stable.

Overall, it's a nice paper. I have one tiny input. In figure 2, lysine residue-thiol need change to lysine residue-amine”.

Answer: This has been done as suggested.

Reviewer 2 Report

This article by Soliman and coworkers presents the conformational dynamics analyses of the HSP72 8-N-benzyladenosine complex, in which it was previously observed (by other authors, Pettinger 2017) the unexpected covalent binding of Lys56, instaed of Cys17. 

The authors report an insight of the binding mode of this interaction, observing that the coupling of 8-N-benzyladenosine to Lysine-56 induces a closed conformation of the IIB and IIA α-helices of the nucleotide-binding subdomain, maintaining the salt-bridge between Glu268 and Lys56, which is a crucial interaction. 

Although I don't feel qualified to judge about the experimental part and the methods used for the analyses, as I am not an expert in computational and theoretical chemistry, I feel that the paper is overall interesting and worthy of publication in Molecules. 

However, the entire manuscript needs an extensive rewriting, as both English and style are negleted. Even in the title, lysine and cysteine are mispelled. The manuscript contains many typos, Figures are often diffucult to read (for example in the caption of Figure 1 color red is mentioned, while in the Figure this color is missing) and many sentences need a substantial rewriting, to make this work publishable.

Author Response

Q1. However, the entire manuscript needs an extensive rewriting, as both English and style are neglected. Even in the title, lysine and cysteine are misspelled. The manuscript contains many typos, Figures are often difficult to read (for example in the caption of Figure 1 color red is mentioned, while in the Figure this color is missing) and many sentences need a substantial rewriting, to make this work publishable.

Answer: Authors have thoroughly read through and all typos and grammatical mistakes have been addressed.

Reviewer 3 Report

In the present manuscript, Aimen Aljoundi, et al. applied covalent MD simulations to HSP72 systems to elucidate the 8-N-benzyladenosine coupling to Lys56 but not Cys17. A series of methods, such as PCA, coordinate analysis, and DSSP, were performed to explore the properties of different systems. Overall, the manuscript presents novel data that can potentially provide new insights into the preference of 8-N-benzyladenosine towards Lys56. However, it is missing a scholarly adequate discussion of the literature and descriptive figures that are important to make the manuscript become a convincing computational report.

The manuscript suffers from the following weaknesses that should be addressed to make it suitable for publication:

  1. the methods of docking. The authors declared their system construction methodology but the way to build the covalent bond is confusing. They firstly used a conventional docking algorithm and manually built the covalent bond. In fact, conventional docking algorithm only thinks of non-covalent interactions. Thus, the output structure of docking can be improper for connect the two bonding atom. Of note, Maestro Schrodinger provides users with a covalent docking module. Considering the current state, the authors should prove the credibility of their method.

  1. data accuracy. In the part of overall structural analysis, it is noticeable that the all data lack of standard deviation, which impacts the credibility of data. For instance, the radius of gyration is extremely similar in the three systems. Considering standard deviation, such data have no statistical significance. It is recommended that the standard deviation is supplied in data analysis.

  1. quality of figure 7. The meaning of different color in figure 7 A and C should be clarified. Also, the axes is not unified, causing it impossible to compare subfigures in figure 7 A and C. In addition, the coordinates of axes are different from each other, hindering the establishment of standards. They should be considered in the revision.

Minor points:

  1. Brackets in line 47 lack of space in front of it.
  2. The abbreviation repeats a lot of times, such as SBD and NBD. It is obvious in line 46-55. ‘N-terminal nucleotide-binding domain (NBD)’ only needs to emerge once.
  3. In line 125, the process of minimization, heating, and equilibrium should be more precise for repeatability.
  4. The overall written language should be checked carefully. For example, in line 149, ‘asses’ should be ‘assess’.
  5. In line 245 and 246, distance values do not need to be as accurate as 5 digits after the decimal point. 9.24 fits the precision of calculation more.
  6. Please describe the biological meaning of the PC1 proportion but not just state the digit.

Author Response

We would like to acknowledge and appreciate your insightful comments and suggestions. We have made a great attempt in addressing each points and concerns raised. Our responses are provided below and all changes highlighted in RED in the revised manuscript.

Comment-1. The methods of docking. The authors declared their system construction methodology but the way to build the covalent bond is confusing. They firstly used a conventional docking algorithm and manually built the covalent bond. In fact, conventional docking algorithm only thinks of non-covalent interactions. Thus, the output structure of docking can be improper for connect the two bonding atom. Of note, Maestro Schrodinger provides users with a covalent docking module. Considering the current state, the authors should prove the credibility of their method.

Response. A detailed description of the docking process has been provided in the revised manuscript (lines 105 - 117)

Comment-2. Data accuracy. In the part of overall structural analysis, it is noticeable that the all data lack of standard deviation, which impacts the credibility of data. For instance, the radius of gyration is extremely similar in the three systems. Considering standard deviation, such data have no statistical significance. It is recommended that the standard deviation is supplied in data analysis.

Response. The data values in the revised manuscript are expressed as mean ± standard deviation (mean ± std) as recommended.

Comment-3. Quality of figure 7. The meaning of different color in figure 7 A and C should be clarified. Also, the axes is not unified, causing it impossible to compare subfigures in figure 7 A and C. In addition, the coordinates of axes are different from each other, hindering the establishment of standards. They should be considered in the revision.

Response. The different colors have been clarified, and axes labeling enhanced. However, a unified axes of the subfigures could not be provided since they are different plots put together based on the range of their different data values.

Comment-4. Brackets in line 47 lack of space in front of it.

Response. This has been amended

Comment-5. The abbreviation repeats a lot of times, such as SBD and NBD. It is obvious in line 46-55. ‘N-terminal nucleotide-binding domain (NBD)’ only needs to emerge once.

Response. This has been amended accordingly.

Comment-6. In line 125, the process of minimization, heating, and equilibrium should be more precise for repeatability.

Response. A detailed account of minimization, heating, and equilibrium process has been provided in lines 133 – 137.

Comment-7. The overall written language should be checked carefully. For example, in line 149, ‘asses’ should be ‘assess.

Response. The Authors have read through, rectify possible typos and grammatical mistakes have been addressed.

Comment-8. In line 245 and 246, distance values do not need to be as accurate as 5 digits after the decimal point. 9.24 fits the precision of calculation more.

Response. This has been corrected as suggested.

Comment-9. Please describe the biological meaning of the PC1 proportion but not just state the digit.

Response. This has accordingly been amended.  

Thank you and Best regards

Authors

Round 2

Reviewer 2 Report

The authors have made the changes recommended and the manuscript is much improved.

Author Response

We greatly appreciate your inputs to our initial submission.

Reviewer 3 Report

The revised manuscript was improved a lot in the description of scientific problems and fits publication better. However, I still have some suggestions for the authors.

  1. With the addition of standard deviation, the differences of ROG is increasingly vague. Thus, I recommend that this part should be deleted before publication.

  1. I do not think the description of the covalent docking process solves my confusion. Please use the covalent docking of Maestro and compare the result with yours. If they are similar enough, it will be convincing that this process works well.

  1. The unit of energy restraint in minimization should be kcal mol−1 Å−2, but not Å.

  1. A time step of 4 fs is not suitable for conventional MD simulation because it has been four times than hydrogen vibration frequency. Please describe the reason why it is applied and the timestep in 250 ns production run.

Author Response

General comments. The revised manuscript was improved a lot in the description of scientific problems and fits publication better. However, I still have some suggestions for the authors.

Response. We greatly appreciate your inputs on our initial submissions.

Comment_1. With the addition of standard deviation, the differences of ROG is increasingly vague. Thus, I recommend that this part should be deleted before publication.

 Response. This has been amended as suggested.

Comment_2. I do not think the description of the covalent docking process solves my confusion. Please use the covalent docking of Maestro and compare the result with yours. If they are similar enough, it will be convincing that this process works well.

 Response. The maestro covdock cysteine-inhibitor covalent complex poses (5) are similar to that generated from our initial protocol. However, the orientation of the maestro covdock lysine-inhibitor complex poses (5) slightly differs from the orientation of our initial lysine-inhibitor complex pose, although they overlap in the same pocket.  

Please find the link below to view the docked complexes from our initial protocol (file name> our_manual ......) and the one generated from the maestro covdock approach (filenames> maestro_covdock.....) for your perusal.

https://drive.google.com/drive/folders/1ytk9LA3Xhs554bXFFDT1Rl9B6cPbSFZL?usp=sharing

Comment_3. The unit of energy restraint in minimization should be kcal mol−1 Å−2, but not Å.

Response. This has been amended as suggested.

Comment_4. A time step of 4 fs is not suitable for conventional MD simulation because it has been four times than hydrogen vibration frequency. Please describe the reason why it is applied and the timestep is 250 ns production run.

Response.  Although a 2-fs integration step would be ideal considering the duration of the simulation (250 ns), however, the use of the 4-fs allowed for more effective use of the limited computational resources while conserving mass and underlying dynamics.

Round 3

Reviewer 3 Report

Thanks for the amendments made by the authors this time. Although the resources can be limited in real work, I strongly recommend that 2fs timestep should be used in the simulation due to the restraint of SHAKE method. 4fs timestep will weaken the conclusion a lot.